# Role of Iron Deficiency in Heart Failure—Clinical and Treatment Approach: An Overview

**DOI:** 10.3390/diagnostics13020304

**Published:** 2023-01-13

**Authors:** Cristina Elena Singer, Corina Maria Vasile, Mihaela Popescu, Alin Iulian Silviu Popescu, Iulia Cristina Marginean, George Alexandru Iacob, Mihai Daniel Popescu, Cristina Maria Marginean

**Affiliations:** 1Department of Mother and Baby, University of Medicine and Pharmacy of Craiova, 200349 Craiova, Romania; 2Department of Pediatric and Adult Congenital Cardiology, Bordeaux University Hospital, 33600 Pessac, France; 3Department of Endocrinology, University of Medicine and Pharmacy of Craiova, 200349 Craiova, Romania; 4Department of Internal Medicine, University of Medicine and Pharmacy of Craiova, 200349 Craiova, Romania; 5Faculty of Medicine, University of Medicine and Pharmacy of Craiova, 200349 Craiova, Romania

**Keywords:** chronic heart failure, iron deficiency anemia, hemoglobin

## Abstract

Background: The association of chronic heart failure (CHF) and iron deficiency (ID) with or without anemia is frequently encountered in current medical practice and has a negative prognostic impact, worsening patients’ exercise capacity and increasing hospitalization costs. Moreover, anemia is common in patients with chronic kidney disease (CKD) and CHF, an association known as cardio-renal anemia syndrome (CRAS) possessing a significantly increased risk of death. Aim: This review aims to provide an illustrative survey on the impact of ID in CHF patients—based on physiopathological traits, clinical features, and the correlation between functional and absolute ID with CHF—and the benefit of iron supplementation in CHF. Method: We selected the most recent publications with important scientific content covering the association of CHF and ID with or without anemia. Discussions: An intricate physiopathological interplay is described in these patients—decrease in erythropoietin levels, activation of the renin-angiotensin-aldosterone system, systemic inflammation, and increases in hepcidin levels. These mechanisms amplify anemia, CHF, and CKD severity and worsen patients’ outcomes. Conclusions: Anemia is frequently encountered in CHF and represents a negative prognostic factor. Data from randomized controlled trials have underlined the administration of intravenous iron therapy (ferric carboxymaltose) as the only viable treatment option, with beneficial effects on quality of life and exercise capacity in patients with ID and systolic heart failure.

## 1. Introduction

Tissue metabolism and many critical biochemical pathways involve the essential trace element iron. The primary consumer of iron is the erythroid bone marrow, which uses it to form new red blood cells. Iron is one of the key elements regarding oxygen homeostasis, including oxygen transport and storage processes [1,2].

Chronic heart failure (CHF) is a commonly encountered disease, with a prevalence of more than 64 million cases worldwide. Patients suffering from this affliction need continuous medical care, including frequent medical monitoring, hospitalizations, and extensive treatment. Thus, CHF is an economic burden for medical systems all around the globe, with a worldwide expenditure predicted to reach USD 400 billion by 2030 [3]. 

CHF is frequently associated with iron deficiency (ID) with or without anemia, both entities representing negative independent predictors. Anemia and ID worsen therapeutical outcomes, increase the need for hospitalization, and decrease the overall quality of life in patients with CHF [4]. The contributions of anemia to heart failure, incompletely elucidated, are multiple and intricate. A series of etiopathogenic hypotheses have been described: iron deficiency, increased levels of AcSDKP (stem cell proliferation inhibitor), excessive secretion of cytokines, hemodilution, and cardiac cachexia.

Moreover, drugs commonly used in the management of CHF, such as angiotensin receptor blockers, angiotensin-converting enzyme inhibitors, anticoagulants, and antiaggregants, pose a risk for anemia development through multiple mechanisms, including direct erythropoiesis inhibition, and gastrointestinal bleeding leading to absolute ID. This fact further underlines the link between anemia and CHF, and the strong likelihood of anemia onset in this group of vulnerable patients.

The association of chronic kidney disease (CKD)—incorporating a decrease in erythropoietin levels—with this condition is defined as cardio-renal anemia syndrome (CRAS). In this pathological entity, the failure of a single organ (heart or kidneys) determines the alteration of the function of the other. The overlap between physiopathological mechanisms creates a vicious cycle in which every dysfunction is perpetually amplified.

The purpose of this review is to provide an illustrative survey on the impact of ID in CHF patients, based on the physiopathology of ID, clinical features of ID and anemia, and the correlation of functional and absolute ID with chronic heart failure, and to analyze the relevance of the most important conducted studies, converging towards the vital benefit of iron supplementation in patients with CHF.

## 2. Iron Deficiency Anemia

Iron deficiency anemia (IDA) is characterized by decreased hemoglobin content in the red blood cells (RBCs) (hypochromia) and diminished mean corpuscular volume (MCV) (microcytosis), both abnormalities being caused by a decrease in iron levels [5]. 

ID is frequently encountered in medical practice. ID can be observed in any geographical area, affecting both genders and all age groups. However, some high-risk groups exist, such as young women, pregnant women, children during growth periods, and the elderly [6].

### 2.1. Physiology and Physiopathology

Iron is mainly present in nutrients, mainly in its ferric form (Fe^3+^). It is reduced by gastric acid and intestinal ferric reductases (i.e., duodenal cytochrome B) to the ferrous form (Fe^2+^). Divalent metal transporter 1 (DMT1) carries the Fe^2+^ ions inside the enterocyte through the cellular membrane. Inorganic iron absorption is facilitated by vitamin C (ascorbic acid), through the improvement of reduction processes, and the formation of soluble complexes. Organic iron is absorbed by the heme carrier protein 1 (HCP-1). Inside the cell, heme is degraded by heme oxygenase-1 (HO-1), releasing ferrous iron [6,7].

Iron can follow two paths inside the enterocyte: being exported into plasma or stored. Intracellular storage of iron uses the protein ferritin, while ferroportin (FPN) exports iron into circulation. The ferroxidase hephaestin converts ferrous iron into ferric iron, which is loaded onto transferrin—the leading plasmatic iron transporter. Besides supplying cells with iron, transferrin also limits toxic radical formation [7,8]. 

Hepcidin is the primary regulator of iron absorption. It is encoded by the HAMP gene on chromosome 19 and synthesized in the liver. By binding to FPN, hepcidin determines its internalization and degradation, limiting intracellular iron’s export into circulation [9].

Generally, a healthy adult with a balanced diet consumes between 3 and 5 g of iron per day. About 65% of all iron in the body is present in red blood cells and in the erythroid bone marrow, which is the primary consumer of iron. Most of the iron used in erythropoiesis is not supplied by the diet, ensuring a daily iron intake of only 1–2 mg, but by splenic recycling of senescent erythrocytes [7,8].

The literature acknowledges two main sub-types of ID: absolute and functional, leading to anemia. Absolute ID represents the depletion of iron stores and occurs in the presence of a decreased intake, impaired absorption, increased demand, or chronic hemorrhage. Functional ID refers to the impaired mobilization of stored iron, secondary to chronic inflammation and raised hepcidin levels, as observed in patients with cancer, obesity, inflammatory bowel disease, chronic kidney disease, and chronic heart failure. Another instance included in functional ID is the imbalance between iron supply and demand when erythropoiesis is accelerated because of elevated erythropoietin (EPO) levels (endogenous response to anemia) or treatment with erythropoiesis-stimulating agents [10].

A hereditary form of ID is described in the literature as iron-refractory iron deficiency anemia (IRIDA), an autosomal recessive inheritance pattern. Matriptase-2 is a protein belonging to the type-2 serine protease family and is encoded by the TMPRSS6 gene. In a healthy human, matriptase-2 diminishes the liver production of hepcidin. Mutations in the TMPRSS6 gene cause a significant rise in hepcidin production, severely reducing iron absorption, and are, therefore, the critical element in IRIDA pathogenesis [11].

### 2.2. Clinical and Biological Features of Iron Deficiency Anemia

Clinically, the evolution of ID can be divided into three distinct stages:Iron depletion without anemia, characterized by exhaustion of iron stores, preserved serum iron, and hemoglobin levels.Moderate normocytic normochromic anemia is preceded by lowering serum iron and increasing total iron-binding capacity (TIBC).Severe microcytic hypochromic anemia, accompanied by characteristic clinical and biological features, facilitates the diagnosis.

An accurate diagnosis of IDA is supported by information provided by history and physical examination, observation of signs and symptoms of IDA (e.g., fatigue, weakness, anorexia, chest pain, headache, pallor, brittle nails, tongue swelling, cold hands and feet, etc.), and laboratory tests.

#### 2.2.1. Peripheral Blood Tests

In IDA, hemoglobin levels measure under 13 g/dL in men and 12 g/dL in women. Hematocrit is also decreased [12].

Microcytosis is characteristic of IDA and hypochromia of RBCs, as shown by values of MCV and mean corpuscular hemoglobin (MCH). However, these changes are encountered late in ID, expressing long-lasting anemia [5]. 

A peripheral blood smear can detect elliptocytosis (cigar-shaped cells), anisocytosis, and anisochromia [13].

The reticulocyte index usually measures under 1%, making IDA hyperproliferative anemia [12]. 

#### 2.2.2. Perls’ Stain

Assessing bone marrow aspirate in Perls’ stain is the gold standard for diagnosing ID. This stain permits direct evaluation of the iron stored as hemosiderin, which is significantly reduced or nil in IDA. The percentage of erythroblasts containing iron-staining granules (sideroblasts) is often reduced to under 10% [14,15].

#### 2.2.3. Serum Iron and Total Iron-Binding Capacity

The serum iron dosage is not an optimal diagnostic test because it shows essential daily variations, is low in inflammatory anemia, and may falsely measure higher levels in iron-deficient patients after oral iron administration. Elevated TIBC is specific for IDA but is relatively insensitive, as it decreases in inflammation and poor nutrition [6].

#### 2.2.4. Transferrin Saturation

Transferrin saturation is defined as the ratio between serum iron and TIBC. Usually, in ID, a transferrin saturation lower than 16% (20% in the presence of inflammation) is detected [10].

### 2.3. Causes of Iron Deficiency

Iron deficiency’s general causes are presented in Table 1.

## 3. Anemia and Iron Deficiency in Heart Failure

Iron deficiency is present in approximately 30% of heart failure patients, usually classified as chronic normocytic anemia. Functional iron deficiency occurs in about one third of patients [16].

In patients with heart failure, the presence of anemia is considered to be a negative prognostic factor. This factor can be corrected to improve the decompensation of cardiac function or the unfavorable evolution of organ failure. Estimates of the prevalence of the association between anemia and heart failure vary widely according to various studies, between 4 and 61%. In a meta-analysis performed by Groenveld et al., reporting on 34 studies published between 2001 and 2007, the prevalence of anemia was estimated at 37.2% (10–49%); a similar percentage was highlighted in the study of anemia in the population with heart failure (STAMINA-HFP), which recorded a prevalence of anemia in 34% of the studied patients [17,18]. Anemia contributes to the decreased exercise capacity of the affected patients, thus increasing hospitalization costs, and it is generally associated with reduced survival [19,20]. 

Table 2 presents the features of heart failure patients with and without anemia, based on insight from three extensive studies, conducted by Tymińska et al., McCullough et al., and Maggioni et al., which provide data regarding the prevalence of ID in CHF patients, and associated comorbidities [21,22,23].

Patients with heart disease and advanced age, female gender, the association of chronic kidney disease, a low body mass index, increased jugular vein pressure, and higher levels of N-terminal proB natriuretic peptide (NT-proBNP) and C-reactive protein (CRP) are considered to have an increased risk of anemia. However, a direct link between the risk factors and the occurrence of anemia has not been demonstrated [24,25,26].

### 3.1. Causes and Etiopathogenic Mechanisms of Iron Deficiency and Anemia in Chronic Heart Failure

In establishing the diagnosis, it is necessary to determine whether anemia is a factor determining heart failure or is only associated with it. The causes of anemia in CHF, incompletely elucidated, are multiple and, of course, intricate. A series of etiopathogenic hypotheses have been described: iron deficiency, excessive secretion of cytokines, hemodilution by sodium retention, cardiac cachexia, the use of angiotensin II converting enzyme inhibitor (ACEI) drugs, and chronic kidney pathology associated with decreased levels of EPO, which characterizes the anemic cardio-renal syndrome, in which failure of a single organ (heart or kidneys) determines the alteration of the function of the other [27,28].

#### 3.1.1. Absolute Iron Deficiency

Absolute ID in CHF can result from a lack of appetite, poor nutrition, gastrointestinal decrease in iron absorption, and gastrointestinal blood loss due to the use of antiaggregant and anticoagulant medication [29,30].

CHF can determine changes in the intestinal wall, including bowel edema and consecutive enterocyte dysfunction, followed by iron malabsorption. Therefore, myocardial dysfunction contributes directly to ID development and anemia [31].

An important number of CHF cases present cardiovascular comorbidities requiring antiplatelet and anticoagulant therapy. These therapeutical agents can induce changes in the digestive mucosa, causing chronic blood loss (Figure 1) [32].

#### 3.1.2. Functional Iron Deficiency

Heart failure has been associated with a systemic pro-inflammatory status that leads to functional ID. The pro-inflammatory group and the association of bacterial proliferation favor the sequestration of iron in stores by increasing the level of hepcidin, thus decreasing the iron availability for hematopoiesis [9,31].

#### 3.1.3. Cardio-Renal Anemia Syndrome

Anemia is a common condition in both CHF and CKD. The three entities form a pathological triangle, potentiating the other entities and aggravating patient outcomes (Figure 2) [33].

The renal synthesis of EPO is stimulated by tissue hypoxia. In CHF, plasma levels of EPO increase when the hemoglobin value decreases, but not proportionally. It has also been highlighted that the bone marrow develops a certain degree of resistance to the action of EPO. However, the most critical alteration in EPO’s activity is CKD, an affliction frequently associated with CHF. Long-term renal ischemia and CKD directly reduce the secretion of EPO [34]. According to Luthi et al., approximately 1 in 5 patients with CHF have CKD, and the prevalence of anemia increases with the decrease in the glomerular filtration rate [35].

The decrease in serum levels of EPO in CHF is also correlated with the excessive secretion of cytokines (TNF alpha, IL6). These cytokines interfere with EPO’s action, reducing iron availability by inhibiting absorption at the gastrointestinal level and blocking it in the reticuloendothelial system [36].

ACE inhibitors and angiotensin receptor blockers (ARBs), therapeutical agents often used in CHF treatment, inhibit erythroid precursors [37]. One of the ACEI mechanisms of action contributing to anemia is mediated by N-Acetyl-Seryl-Aspartyl-Lysyl-Proline (AcSDKP), a tetrapeptide with an antiproliferative effect that acts by inhibiting erythropoiesis. The action of the converting enzyme normally metabolizes AcSDKP. By using ACEIs, the enzyme is inhibited, thus increasing AcSDKP activity and inhibiting erythropoiesis. Increased serum levels of AcSDKP were highlighted in patients on ACEI treatment, along with a reduced level of ACE activity, which supports this pathogenetic hypothesis (Figure 1) [28].

The effect of ACEIs on erythrocyte formation in patients with heart failure was evaluated in clinical trials, which compared randomized groups using enalapril and placebo, respectively, with the results supporting a higher incidence of anemia in patients being treated with seemingly a dose-dependent effect [28,38].

Angiotensin II is responsible for vasoconstriction as a direct response to the hemodynamic modifications generated by CHF. A decrease is followed by vasoconstriction in renal perfusion and increases in renin production. Renin transforms angiotensinogen into angiotensin, thus perpetuating a vicious circle and accentuating organ dysfunction. The renin-angiotensin-aldosterone system generates hemodilution through hydrosaline retention, another known causal factor of anemia in patients with heart failure [39].

Renal hypoperfusion caused by CHF determines kidney injury. Decreases in cardiac output, renal blood flow, efferent arteriole vasoconstriction, and neurohormonal activation are followed by changes in intrarenal circulation and increased sodium and water reabsorption. Central venous pressure and its expression in the renal veins lead to a decrease in the glomerular filtration rate, with consequent worsening of renal dysfunction. Increased levels of renin, angiotensin, and aldosterone cause damage directly to cardiac cells, thus exacerbating existing damage. Low renal blood flow also causes an increase in antidiuretic hormone secretion, exacerbating renal vasoconstriction and sodium and water retention [35].

### 3.2. Diagnosis of Iron Deficiency 

There is a relatively high rate of underdiagnosis of ID because anemia is often not clinically evident. Furthermore, anemia may be completely absent, as it needs a specific time frame to develop after the appearance of ID. In a study by Martens, ID was present in 53% of patients and more frequent in patients with reduced ejection fraction. At the same time, anemia was present in a smaller number of cases: 36% [40].

Diagnostic criteria for ID in CHF patients, with or without anemia, refer to serum ferritin <100 µg/L, or ferritin between 100 µg/L and 299 µg/L, and transferrin saturation <20% [41].

Routinely screening for ID in the entire population of patients with cardiac disease is highly recommended, considering ID’s prevalence among the general demographic and its significant implications for the prognostics of a patient suffering from CHF [42].

### 3.3. Clinical Consequences of Iron Deficiency in Heart Failure

According to the literature, ID has been correlated with various clinical consequences only in the presence of anemia. However, recent studies have shown that ID may be an independent predictor of adverse outcomes in patients with systolic heart failure [26]. Moreover, ID with or without clinically evident anemia has been associated with a deterioration of physical performance and exacerbation of dyspnea and fatigue [43].

ID caused an increase in the length of hospitalization in patients with heart failure [25]. The association of ID with heart failure also determined a significant decrease in the health-related quality of life (HRQL), assessed by the Minnesota standardized questionnaire. A European multi-observational study of 1278 heart failure patients showed that iron deficiency affected the quality of life and human life, independent of anemia. In this study, 58% of patients were diagnosed with ID, and 35% had anemia. HRQL in patients with iron deficiency and anemia was lower than in those with heart failure but without iron deficiency or anemia [44].

ID also has a negative impact on exercise capacity in heart failure patients, based on the fact that non-hematopoietic tissues, including skeletal and cardiac muscle tissue, depend on the presence of iron as a critical element in the constitution of proteins involved in vital cellular processes, namely oxygen storage (a component of myoglobin) and oxidative energy metabolism methods (part of oxidative enzymes) [45,46,47].

ID has been highlighted in clinical trials as an independent factor correlated with increased morbidity and mortality in patients diagnosed with heart failure without the mandatory association of anemia [43,48].

### 3.4. Management of Iron Deficiency in Heart Failure

#### 3.4.1. Intravenous Iron Supplementation Therapy

The 2016 European Society of Cardiology (ESC) guidelines for heart failure address the clear benefit of treating ID in patients with heart failure, where ID is defined as an essential entity independent of the presence of anemia [49].

Treatment options for correcting iron deficiency in the general population are intravenous (IV) or oral iron. In patients with heart failure with low ejection fraction (LVEF), however, the 2016 ESC guidelines specifically recommended that ID should be treated by prescribing iron compounds with parenteral administration, such as IV ferric carboxymaltose (Figure 3), as oral iron therapy is ineffective for replenishing iron stores (Table 3) [50].

The efficacy of ferric carboxymaltose has been evaluated in several randomized clinical trials. In 2012, Avni et al. [51] published a meta-analysis including several controlled studies [25,52,53] in which the effectiveness of intravenous iron administration was compared in 370 patients versus 224 controls, with noticeable favorable results in the treated group of patients, regarding improvement of quality of life, NYHA class improvement, decrease in CRP level, and reducing the need for hospitalization.

Another meta-analysis by Qian et al., which included studies on 907 patients who received intravenous iron-ferric carboxymaltose treatment, confirmed a decrease in the number of hospitalizations of CHF patients [54].

In 2015, the multicenter placebo-controlled CONFIRM-HF study, which enrolled 304 symptomatic ambulatory HF patients with LVEF ≤45% and ID and is the study with the most extended published follow-up period in patients treated with iron (52 weeks), revealed significant NYHA class improvements and quality of life improvement, statistically reported after 24 weeks of treatment. At the same time, IV iron treatment reduced the need for hospitalization in the studied patients [55].

In 2015, a study performed by Toblli et al., after six months of IV iron administration, revealed an improvement in the function of the left ventricle correlated with the reduction in ID, the decrease in inflammatory markers, and the remission of symptoms, as well as the improvement of kidney function, with an increase in glomerular filtration rate [56].

A recent 2018 meta-analysis of individual patient data by Anker et al. proved a reduction in recurrent hospitalizations and cardiovascular mortality in patients with CHF and ID treated with IV ferric carboxymaltose [57].

However, these clinical trials did not evaluate the effect of treatment on survival, so, to date, the potential efficacy of IV iron therapy has not been evaluated in terms of mortality reduction in patients with CHF. Moreover, the effectiveness and safety of intravenous ferric carboxymaltose treatment have not been established yet in acute heart failure or heart failure with preserved ejection fraction (LVEF ≥ 50%), requiring further investigation in more extensive future studies [58,59].

Although these findings related to iron carboxymaltose therapy should be confirmed by extensive clinical trials, diagnosis and treatment of ID to improve exercise capacity, symptoms, and quality of life will remain an essential part of an optimal approach to CHF.

The use of IV ferric carboxymaltose in patients with ID and CHF is restricted in patients with contraindications, such as hypersensitivity to the active substance or any of its excipients, diagnosed severe hypersensitivity to other IV iron compounds, presence of anemia with causes other than ID (for example, other microcytic anemias), iron overload, or iron metabolism disorders.

#### 3.4.2. Oral Iron Supplementation Therapy

Oral iron supplementation therapy offers some practical advantages, which are limited where absorption in the gastrointestinal tract is reduced. The IRONOUT-HF study concluded that there was a minimal increase in iron stores in CHF patients with reduced LVEF who followed oral iron therapy. Additionally, no improvement in exercise capacity was recorded [50]. Moreover, oral iron is poorly tolerated in patients with CHF, who may present digestive symptoms and gastrointestinal side effects in up to 60% of cases. Oral iron is less effective than IV iron, and a relatively long duration of oral iron therapy is always necessary (in some cases > 6 months) to refill iron stores, thus being a therapy often limited by gastrointestinal adverse effects [46].

#### 3.4.3. Blood Transfusion

Red blood cell transfusion is recommended under careful monitoring in cases of severe anemia (hemoglobin value below 7 g/dL) [28]. The therapy must be strictly individualized considering the cost–benefit ratio because the procedure presents risks and possible immunological (hemolysis, hyperthermia, allergic reactions, purpura, etc.) and non-immunological (fluid overload, electrolyte disturbances, infections) complications.

#### 3.4.4. Erythropoiesis-Stimulating Agents

According to several authors, the ideal treatment should include the combination of erythropoiesis-stimulating agents with iron therapy, improving symptoms, decreasing hospitalizations, and increasing exercise capacity and quality of life [60].

The use of erythropoietin in treating moderate anemia associated with heart failure can beneficially affect cardiac function, improving both NYHA functional class and LVEF, decreasing atrial natriuretic peptide values, and improving kidney function. However, recombinant human erythropoietin is associated with unfavorable side effects: increased blood pressure due to increased hematocrit and micro vascularization damage, risk of deep vein thrombosis due to increased blood viscosity, and risk of aplastic anemia secondary to the production of anti-rHuEpo antibodies [61].

A human EPO analog, darbepoetin, may also be used, with proven clinical benefits such as exercise capacity and quality of life improvement. However, a randomized, double-blind trial conducted in 2008 (STAMINA-HeFT) claims that darbepoetin alfa treatment, administered subcutaneously every two weeks for 12 months, is well tolerated and increases hemoglobin, while associated morbidity and mortality are lowered [62].

Other studies have stated that current clinical trials of erythropoiesis-stimulating agents have failed to provide convincing evidence of decreased morbidity and mortality or long-term effects, thus highlighting the importance of ID, iron metabolism modification, and hemodilution, with an increased risk of administration due to the high incidence of thromboembolic complications [63]. 

#### 3.4.5. Developing Strategies

Hepcidin levels may be normalized by decreasing its production or antagonizing the ferroportin internalization and degradation effect, increasing ferritin levels [64].

## 4. Iron Deficiency’s Impact on Other Cardiovascular Diseases

ID was observed in approximately 50% of patients with acute decompensated HF, being associated with a longer duration of hospitalization [65,66].

In patients with coronary artery disease (CAD) or acute coronary syndrome (ACS), anemia and ID are commonly encountered, and their appearance is correlated with the disease’s severity. This association was documented in a study on 2082 patients conducted by Lanser et al., which revealed the relationship between the presence of anemia and the severity of cardiovascular disease (reflected by an increased rate of stenosis on coronary angiography, NYHA class progression, and LVEF reduction). There was an increased level of pro-inflammatory markers, all correlated with an increased risk of cardiovascular events, which could be mainly due to the underlying inflammation. The inflammation incriminated may lead to, and be caused by, atherosclerosis [67].

In patients hospitalized for ACS, the prevalence of ID varied between 29 and 56%, depending on the characteristics of the study groups [68,69]. A study by Zeller et al., which included 836 patients suffering from ACS, showed evidence of a 50% increased risk of non-fatal myocardial infarction and cardiovascular death in patients with ID or anemia [68].

Data on the link between ID and the risk of cerebrovascular disease are sparse and conflicting. The prevalence of ID in stroke patients could be around 45%. A Mendelian randomized trial showed an increased risk of stroke, particularly cardioembolic stroke, in patients with higher serum iron and ferritin or TSAT but lower serum transferrin concentrations [70]. In contrast, in another population-based study that enrolled people aged ≥ 65 years, low serum iron concentrations were associated with an increased risk of stroke [71].

The prognostic significance of ID in cerebrovascular diseases has not been quantified, and, consequently, the prognostic impact of treating ID in these patients is not sufficiently evaluated. However, ID may be associated with a severe clinical status.

In a study by Kvaslerud et al., which analyzed 464 patients with aortic valve stenosis, 53% had ID and 20% had anemia. Although patients with ID had a severe clinical profile, no association between ID, TSAT, or hepcidin level and the risk of major adverse cardiovascular events or mortality was demonstrated, regardless of the therapeutic approach of aortic stenosis [72]. Thus, although ID could be considered a negative prognostic marker in patients with severe aortic stenosis, its determining role on the evolution of the condition is limited.

The prognostic role of ID in atrial fibrillation (AF) is unknown [73]. ID prevalence was approximately 50% in patients with permanent AF, compared to 30% in patients with persistent AF. The benefit of ID therapy has yet to be evaluated, requiring further studies [74].

Although numerous studies highlight the potential role of ID association with cardiovascular diseases, such as CAD, aortic stenosis, cerebrovascular disease, and AF, the evidence is contradictory, and the underlying pathophysiological mechanisms still need to be completed. Future studies are required to complete the characterization of patients with cardiovascular disease and identify patients who will benefit from iron supplementation therapy [65].

## 5. Conclusions

Anemia is frequently encountered in CHF, being a negative prognostic factor. The causes of anemia, incompletely elucidated, are varied and challenging.

Screening for ID in patients with cardiac disease is highly recommended, considering its relevance for the prognostics of these patients.

Current guidelines recommend only one intravenous iron preparation in CHF—ferric carboxymaltose. The literature has established its beneficial effect, with improvements in quality of life and exercise capacity.

The administration of oral iron supplements in patients with CHF is not recommended because of the lack of studies demonstrating its efficacy, the high incidence of adverse digestive effects, and the long treatment periods needed to increase hemoglobin levels.

The management of these cases should be handled by a multidisciplinary team, which will collaborate in performing an active screening for diagnosis, prevention, and management to reduce morbidity and mortality from cardiac causes.

## Figures and Tables

**Figure 1 diagnostics-13-00304-f001:**
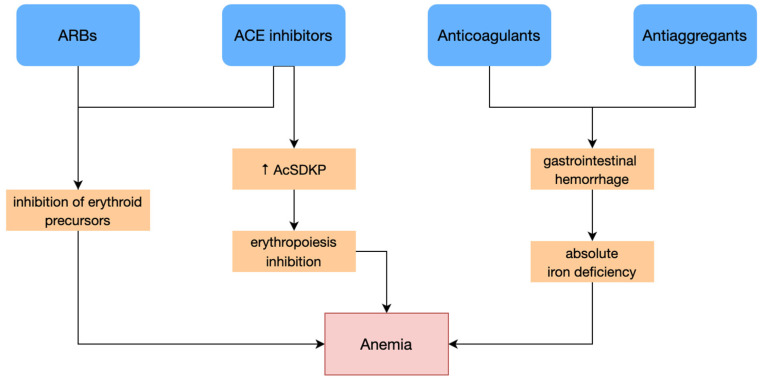
Therapeutical agents used in CHF management and their pathogenic mechanisms. ARB, angiotensin receptor blockers. ACE, angiotensin-converting enzyme. AcSDKP, N-Acetyl-Seryl-Aspartyl-Lysyl-Proline. ↑ = elevated value.

**Figure 2 diagnostics-13-00304-f002:**
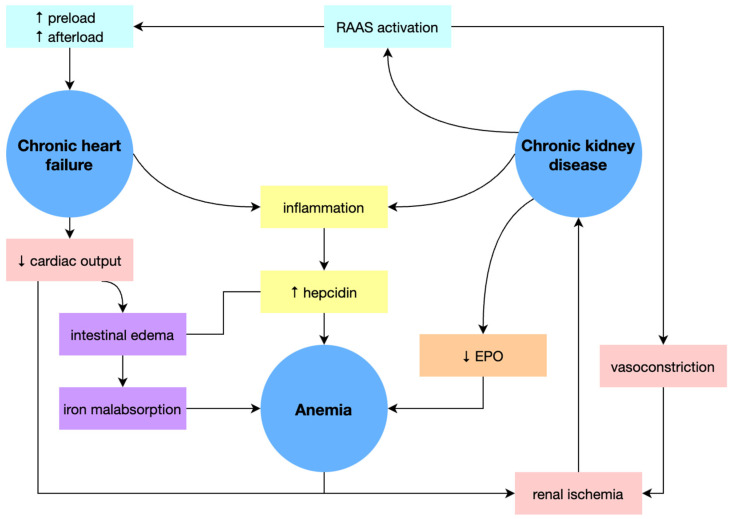
Physiopathology of cardio-renal anemia syndrome. EPO, erythropoietin. RAAS, renin-angiotensin-aldosterone system. ↑ = elevated value, ↓ = decreased value.

**Figure 3 diagnostics-13-00304-f003:**
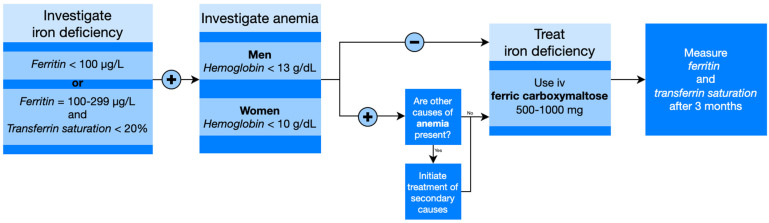
Therapeutical management of iron deficiency.

**Table 1 diagnostics-13-00304-t001:** Possible causes for iron deficiency anemia NSAID, a nonsteroidal anti-inflammatory drug. CHF, chronic heart failure. CKD, chronic kidney disease.

Causes of Iron Deficiency Anemia	
**1. Anemia due to martial deficiency**	
(a) insufficient reserves:	prematurity, twinship, neonatal hemorrhages, maternal anemia
(b) insufficient food intake:	diet with excess flour, exclusive diet with goat’s milk, protein and vitamin deficiencies, vegetarian diet
(c) deficient absorption:	presence of inhibitory factors (phytate, phosphates, carbonates), lack of reducing factors (vitamin C, hydrochloric acid, bile acids), celiac disease, gastrectomy, Helicobacter Pylori infection, intestinal resections, bacterial overgrowths
**2. Iron-loss anemia**	
(a) gastro-intestinal hemorrhages:	esophageal varices (liver cirrhosis), diaphragmatic hernia, esophagitis, gastro-duodenal ulcer, cancer of the digestive tract (esophageal, gastric, colonic cancer), tumors of the small intestine, Vaterian ampulloma, hemorrhoids, rectal polyps, intestinal parasites, celiac disease, Crohn’s disease, ulcerative colitis, colonic angiodysplasia, bariatric surgery, NSAIDs consumption
(b) hemorrhages of respiratory origin:	epistaxis, pulmonary tuberculosis, lung cancer, bronchiectasis, pulmonary microinfarcts, alveolar hemorrhage
(c) genito-urinary hemorrhages:	prolonged menstrual cycle, metrorrhagia, renal tuberculosis, renovesical cancer, hemorrhagic nephritis, hemodialysis
(d) hemorrhagic diathesis:	alteration of the capillary wall, alteration of platelets, combined alterations
(f) hypersplenism:	
(g) genetic causes:	iron-refractory iron deficiency anemia
(h) mechanical fragmentation of RBCs:	prosthetic valves
(i) endocrine diseases:	hypothyroidism, pituitary insufficiency, autoimmune polyglandular syndromes
(j) autoimmune diseases:	scleroderma, rheumatoid arthritis, lupus
(k) drugs:	anticoagulants, antiaggregants, NSAIDs
(l) CHF, CKD.	

**Table 2 diagnostics-13-00304-t002:** A. Prevalence of anemia among heart failure patients. B. Features of heart failure patients with and without anemia.

**A.**																		
	Total	Anemic																
Tymińska et al. [21]	1394	466 (33.42%)																
McCullough et al. [22]	1763	728 (41.29%)																
Maggioni et al. [23]	7421	828 (11.15%)																
**B.**																		
	**Tymińska et al. [21]**						**McCullough et al. [22]**						**Maggioni et al. [23]**					
	Total		Non-anemic		Anemic		Total		Non-anemic		Anemic		Total		Non-anemic		Anemic	
Number of patients	1394		928		466		1763		1035		728		7421		6593		828	
Men	918	(65.85%)	599	(64.55%)	319	(68.45%)	1283	(72.77%)	847	(81.84%)	436	(59.89%)	5887	(79.33%)	5246	(79.57%)	641	(77.42%)
Hypertension	959	(68.79%)	629	(67.78%)	330	(70.82%)	1068	(60.58%)	605	(58.45%)	463	(63.60%)	—		—		—	
CKD	293	(21.02%)	161	(17.35%)	132	(28.33%)	—		—		—		—		—		—	
COPD	261	(18.72%)	142	(15.30%)	119	(25.54%)	159	(9.02%)	79	(7.63%)	80	(10.99%)	—		—		—	
NYHA class III/IV	1032	(74.0%)	649	(69.94%)	383	(82.19%)	644	(36.53%)	337	(32.56%)	307	(42.17%)	2698	(36.36%)	2279	(34.57%)	419	(50.60%)

CKD, chronic kidney disease. COPD, chronic obstructive pulmonary disease. NYHA, New York Heart Association.

**Table 3 diagnostics-13-00304-t003:** Comparison of the efficacy and side effects of ferric carboxymaltose versus oral iron supplements.

Ferric Carboxymaltose	Parameter	Oral Iron Supplements
Significant	↑ Hemoglobin levels	Minimal
+	NYHA class improvement	−
+	HRQL improvement	−
+	↑ Exercise capacity	−
+/−	Gastro-intestinal adverse effects	+

HRQL, health-related quality of life. ↑ high value, + present, − absent.

## Data Availability

Not applicable.

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
