# Peer review of "Role of Iron Deficiency in Heart Failure—Clinical and Treatment Approach: An Overview"

_diagnostics, 2023, doi:10.3390/diagnostics13020304_

Round 1

Reviewer 1 Report

Below are my views on the article sent to me for consideration:

Heart failure is an important cause of morbidity and mortality globally. The incidence of heart failure 1–2% in the over-all population and as much as 10% in the population over 65 years of age. Iron deficiency is common in patients with heart disease. Over half of patients with coronary artery disease, and much of those with heart failure (HF) have iron deficiency.

For this reason, I find this review, which deals with iron deficiency in heart failure patients in detail, very valuable. In order to make the review more useful, it would be good to show the large randomized controlled studies on this subject in a table comparatively. It would also be good to mention the association of iron deficiency with conditions such as acute decompensation, coronary artery disease, cerebrovascular disease, aortic stenosis and atrial fibrillation, which can be seen in heart failure patients. Spelling and grammatical errors in the article should also be corrected.

Author Response

Dear Reviewer,

Thank you for your appreciation and your constructive suggestions.

Q1.For this reason, I find this review, which deals with iron deficiency in heart failure patients in detail, very valuable. In order to make the review more useful, it would be good to show the large randomized controlled studies on this subject in a table comparatively.

A1.We completed the review as you asked, showing in a comparative table the extensive randomized controlled studies referring at the prevalence of the essential features of heart failure patients with and without anemia.

Anemia represents one of the significant adverse prognostic factors for heart failure. This factor can be corrected to improve the decompensation of cardiac function or the unfavorable evolution of organ failure. Estimates of the prevalence of the association between anemia and heart failure vary widely, according to various studies, between 4-61%. In a meta-analysis performed by Groenveld et al., reporting on 34 studies published between 2001 and 2007, the prevalence of anemia was estimated at 37.2% (10-49%); a similar percentage was highlighted in the study of anemia in the population with heart failure (STAMINA-HFP), which recorded a prevalence of anemia in 34% of the studied patients [17,18]. Anemia contributes to the decreased exercise capacity of the affected patients, thus increasing hospitalization costs, and it is generally associated with reduced survival [19,20].

Table 2 presents the features of heart failure patients with and without anemia, based on insight from three extensive studies, conducted by Tymińska et al., McCullough et al. and Maggioni et al., which provide data regarding the prevalence of ID in CHF patients, and associated comorbidities [21–23].

Table 2. A. Prevalence of anemia among heart failure patients. B. Features of heart failure patients with and without anemia.

A.

Total

Anemic

Tymińska et al.

1394

466 (33.42%)

McCullough et al.

1763

728 (41.29%)

Maggioni et al.

7421

828 (11.15%)

B.

Tymińska et al.

McCullough et al.

Maggioni et al.

Total

Non-anemic

Anemic

Total

Non-anemic

Anemic

Total

Non-anemic

Anemic

Number of patients

1394

928

466

1763

1035

728

7421

6593

828

Men

918

(65.85%)

599

(64.55%)

319

(68.45%)

1283

(72.77%)

847

(81.84%)

436

(59.89%)

5887

(79.33%)

5246

(79.57%)

641

(77.42%)

Hypertension

959

(68.79%)

629

(67.78%)

330

(70.82%)

1068

(60.58%)

605

(58.45%)

463

(63.60%)

CKD

293

(21.02%)

161

(17.35%)

132

(28.33%)

COPD

261

(18.72%)

142

(15.30%)

119

(25.54%)

159

(9.02%)

79

(7.63%)

80

(10.99%)

NYHA class III/IV

1032

(74.0%)

649

(69.94%)

383

(82.19%)

644

(36.53%)

337

(32.56%)

307

(42.17%)

2698

(36.36%)

2279

(34.57%)

419

(50.60%)

CKD, chronic kidney disease. COPD, chronic obstructive pulmonary disease. NYHA, New York Heart Association.

Patients with heart disease and advanced age, female gender, the association of chronic kidney disease, a low body mass index, increased jugular vein pressure, higher levels of N-terminal proB natriuretic peptide (NT-proBNP) and C-reactive protein (CRP) are considered to have an increased risk of anemia. However, a direct link between the risk factors and the occurrence of anemia has not been demonstrated [24–26].

Q2.It would also be good to mention the association of iron deficiency with conditions such as acute decompensation, coronary artery disease, cerebrovascular disease, aortic stenosis and atrial fibrillation, which can be seen in heart failure patients. Spelling and grammatical errors in the article should also be corrected.

A2.In a summary sub-section, we also mentioned the impact of iron deficiency in cardiovascular pathologies, such as acute decompensation of heart failure, coronary artery disease, acute coronary syndrome, cerebrovascular disease, and atrial fibrillation.

4. Iron deficiency’s impact on other cardiovascular diseases

In patients with acute decompensated HF, ID was observed in approximately 50% of patients, being associated with a longer duration of hospitalization [65,66].

In patients with coronary artery disease (CAD) or acute coronary syndrome (ACS), anemia and ID are commonly encountered, and their appearance is correlated with the disease's severity. This association was documented in a study on 2082 patients conducted by Lanser et al., which revealed the relationship between the presence of anemia and the severity of cardiovascular disease (reflected by an increased rate of stenosis on coronary angiography, NYHA class progression, and LVEF reduction), an increased level of pro-inflammatory markers, all correlated with an increased risk of cardiovascular events, which could be mainly due to the underlying inflammation. The inflammation incriminated may lead to, and be caused by, atherosclerosis [67].

In patients hospitalized for ACS, the prevalence of ID varied between 29-56%, depending on the characteristics of the study groups [68,69]. The study by Zeller et al., which included 836 patients suffering from ACS, evidentiated a 50% increased risk of non-fatal myocardial infarction and cardiovascular death in patients with ID or anemia [68].

Data on the link between ID and the risk of cerebrovascular disease are sparse and conflicting. The prevalence of ID in stroke patients could be around 45%. A Mendelian randomized trial showed an increased risk of stroke, particularly cardioembolic stroke, in patients with higher serum iron and ferritin or TSAT but lower serum transferrin concentrations [70]. In contrast, in another population-based study that enrolled people aged ≥ 65 years, low serum iron concentrations were associated with an increased risk of stroke [71].

The prognostic significance of ID in cerebrovascular diseases has not been quantified, and consequently the prognostic impact of treating ID in these patients is not sufficiently evaluated, however ID may be associated with a severe clinical status.

In a study by Kvaslerud et al., which analyzed 464 patients with aortic valve stenosis, 53% had ID and 20% had anemia. Although patients with ID had a severe clinical profile, no association between ID, TSAT, or hepcidin level and the risk of major adverse cardiovascular events or mortality was demonstrated, regardless of the therapeutic approach of aortic stenosis [72]. Thus, although ID could be considered a negative prognostic marker in patients with severe aortic stenosis, its determining role on the evolution of the condition is limited.

The prognostic role of ID in atrial fibrillation (AF) is unknown [73]. ID prevalence was approximately 50% in patients with permanent AF compared to 30% in patients with persistent AF. The benefit of ID therapy has yet to be evaluated, requiring further studies [74].

Although numerous studies highlight the potential role of ID association with cardiovascular diseases, such as CAD, aortic stenosis, cerebrovascular disease, AF, the evidence is contradictory, and the underlying pathophysiological mechanisms still need to be completed. Future studies are required to complete the characterization of patients with cardiovascular disease and identify patients who will benefit from iron supplementation therapy [65].

Additionally, we performed an English revision to better understand our manuscript.

Kind regards,

Corina Vasile

Reviewer 2 Report

This manuscript is written well. However, moderate proofreading is required. Some sentences are too long. In addition, the abstract should be shorter and follow a structure (background, aim, method, discussion, and conclusion). The introduction was too short and did not mention any rationale for conducting the review. All old references have to be replaced with updated ones.

Author Response

`Dear Reviewer,

Thank you for your appreciation and consideration.

Q1.However, moderate proofreading is required. Some sentences are too long. In addition, the abstract should be shorter and follow a structure (background, aim, method, discussion, and conclusion).

A2. We performed an English revision and rephrased some of our sentences.

Q2.The introduction was too short and did not mention any rationale for conducting the review.

A2. We added more information in the introduction section and the rationale of our review.

The causes of anemia in heart failure, incompletely elucidated, are multiple and intricated. A series of etiopathogenic hypotheses have been described: iron deficiency; increased levels of AcSDKP (stem cell proliferation inhibitor), excessive secretion of cytokines, hemodilution, cardiac cachexia, angiotensin II converting enzyme inhibitors use, and chronic kidney disease associated with decreased levels of erythropoietin, defining the anemic cardio-renal syndrome (CRAS), in which failure of a single organ (heart or kidneys) determines the alteration of the function of the other.

The purpose of this review is to provide an illustrative survey on the impact of ID in CHF patients, based on physiopathology of ID, clinical features of ID and anemia and the correlation between functional and absolute ID with chronic heart failure, and to analyze the relevance of the most important conducted studies that converged towards the vital benefit of iron supplementation in patients with CHF.

Q3.All old references have to be replaced with updated ones.

A3. We updated our references list with more recent titles.

  1. Ford J. Red blood cell morphology. Int J Lab Hematol. 2013 35(3):351-7, doi: 10.1111/ijlh.12082.
  2. Stuart-Smith SE, Hughes DA, Bain BJ. Are routine iron stains on bone marrow trephine biopsy specimens necessary? J Clin Pathol. 2005, 58(3):269-72. doi: 10.1136/jcp.2004.017038.

Kind regards,

Corina Vasile

Round 2

Reviewer 2 Report

The manuscript did not revise well!

1. It has been requested to modify the abstract in a shorter format  (background, aim, method, discussion, and conclusion) with keypoints of review; however, it did not improve well!

2. The introduction also did not improve according to the expectation; thus another round is necessary for revision.

3. The conclusion should be shorter with general keypoints of review!

4. A native proofreader is required to modify the manuscript!

5. Some old references such as no. 38 & 63 still are remained, which should be removed!

Author Response

Dear reviewer,

Thank you once again for taking the time to review our work, and we apologize for not completing all of your requirements:

  1. We have shortened the abstract according to your requirements( background, aim, methods, discussion, and conclusion).
  2. The introduction has been revised as well.
  3. A native speaker proofreader has revised our manuscript.
  4. References 38 and 63 have been replaced with more actual ones.

Round 3

Reviewer 2 Report

The manuscript has been improved but extensive proofreading is required.